# Long-Term Survival After Thyroidectomy for Thyroid Cancer: A Propensity-Matched TriNetX Study with Specialty-Stratified Analyses

**DOI:** 10.3390/cancers17183051

**Published:** 2025-09-18

**Authors:** Ci-Wen Luo, Meng-Hao Chang, Lan Lin, Frank Cheau-Feng Lin, Shih-Wei Chen, Yu-Hsiang Kuan, Pei-Chi Tsai, Ji-Kuen Yu, Stella Chin-Shaw Tsai

**Affiliations:** 1Department of Medical Research, Tungs’ Taichung MetroHarbor Hospital, Taichung 435, Taiwan; kkjj88440@gmail.com; 2Department of Nursing, Jen-Teh Junior College of Medicine, Nursing and Management, Miaoli 356, Taiwan; 3Department of Surgery, Tungs’ Taichung MetroHarbor Hospital, Taichung 432, Taiwan; gaborchang@gmail.com; 4Department of Surgery, National Yang-Ming University Hospital, Yilan 260, Taiwan; 18018@hosp.nycu.edu.tw; 5Department of Surgery, Chung Shan Medical University Hospital, Taichung 402, Taiwan; tpn@csmu.edu.tw; 6School of Medicine, Chung Shan Medical University, Taichung 402, Taiwan; 7Department of Otolaryngology, Tungs’ Taichung MetroHarbor Hospital, Taichung 435, Taiwan; t7819@ms.sltung.com.tw; 8Department of Pharmacology, School of Medicine, Chung Shan Medical University, Taichung 402, Taiwan; kuanyh@csmu.edu.tw (Y.-H.K.); 14614@ms3.sltung.com.tw (P.-C.T.); 9Department of Pharmacy, Chung Shan Medical University Hospital, Taichung 402, Taiwan; 10Master’s Program, Department of Information Management, National Chung Hsing University, Taichung 402, Taiwan; 11Department of Post-Baccalaureate Medicine, College of Medicine, National Chung Hsing University, Taichung 402, Taiwan; 12College of Life Sciences, National Chung Hsing University, Taichung 402, Taiwan

**Keywords:** thyroidectomy, surgical specialty, general surgery, otolaryngology, survival analysis, propensity score matching, TriNetX, endocrine oncology

## Abstract

**Simple Summary:**

Thyroid cancer is the most common endocrine malignancy, and surgery is a key component of care when appropriate. In this study, we asked two practical questions: Do patients live longer when they undergo surgery, and do outcomes vary by the operating specialty (otolaryngology/ENT vs. general/endocrine surgery)? Using a large US electronic health record network (TriNetX), we created comparable groups of patients who did or did not undergo thyroidectomy and evaluated their long-term survival. Patients who had surgery lived longer than similar patients managed without surgery, with a 31.5% relative reduction in the hazard of death among the surgical patients. When we examined the outcomes by specialty, differences were observed; however, these analyses were exploratory and may reflect differences in tumor features, referral patterns, and surgeon or center experience that were not fully captured. These findings support timely surgery within multidisciplinary pathways and highlight the need for prospective studies to understand why outcomes may vary across specialties.

**Abstract:**

Background/Objectives: Whether thyroidectomy confers a long-term survival advantage over non-surgical management in real-world practice remains uncertain. We primarily evaluated the association between surgery and all-cause mortality in thyroid cancer; specialty-stratified outcomes were prespecified as secondary, exploratory analyses. Methods: Using the TriNetX US Collaborative Network (2008–2024), we identified adults with thyroid cancer and created 1:1 propensity score-matched cohorts of patients who did or did not undergo thyroidectomy, balancing demographics, comorbidities, medications, and laboratory variables. Overall survival was assessed with Kaplan–Meier curves and Cox proportional hazard models. Among the surgical patients, we performed exploratory analyses stratified by operating specialty (otolaryngology–head and neck surgery (reference) vs. general/endocrine surgery and other/unknown, reported descriptively). Results: After matching, 49,219 patients were included per cohort. Thyroidectomy was associated with lower long-term mortality versus non-surgical care (adjusted HR 0.685, 95% CI 0.652–0.721). Among the surgical patients, secondary, exploratory specialty-stratified analyses suggested differences: compared with otolaryngology–head and neck surgery (ENT–HNS; reference), general/endocrine surgery (GS/ES) had a lower adjusted hazard of death (aHR 0.561, 95% CI 0.481–0.654), whereas other/unknown specialties had a higher adjusted hazard (aHR 1.583, 95% CI 1.302–1.924). These patterns are hypothesis-generating and may reflect residual confounding, including the tumor stage and histology, referral pathways, and surgeon or center experience. Conclusions: In a large, propensity-matched real-world cohort, surgery was linked to improved long-term survival regarding thyroid cancer. Observed specialty-related variation should be interpreted cautiously, and prospective studies incorporating tumor-level variables and provider/center characteristics are needed. Emphasis should remain on timely surgery within multidisciplinary care pathways.

## 1. Introduction

Thyroid cancer is the most prevalent endocrine malignancy, with a steadily increasing global incidence attributed to the widespread use of diagnostic imaging and heightened disease awareness [1,2]. While the majority of thyroid cancers, particularly differentiated types such as papillary thyroid carcinoma, have an indolent course and favorable prognosis [3,4], surgical intervention remains a key component of management when indicated. Total or subtotal thyroidectomy is commonly warranted for patients with malignancy, compressive symptoms, or functional thyroid disease refractory to medical therapy [5,6]. However, despite generally excellent outcomes, a central question that remains to be answered is whether undergoing thyroidectomy, compared with non-surgical management, is associated with improved long-term survival in real-world settings [7,8]. Differences by operating specialty may exist but are less certain and potentially confounded. Given the complexity of the cervical anatomy and the need to preserve vital structures, such as the recurrent laryngeal nerve and parathyroid glands, surgical expertise is important [9]. Surgeons from various disciplines, including general surgery (GS), otolaryngology–head and neck surgery (ENT), thoracic surgery, and plastic surgery, often perform thyroid procedures depending on institutional norms and regional referral patterns [10,11]. Prior studies have emphasized the volume–outcome relationship in thyroid surgery [12,13,14], but the association between surgical specialty and long-term survival is not well characterized and should be interpreted as exploratory.

Recent findings from nationwide datasets, including the Taiwan National Health Insurance Research Database, suggest that surgeries performed by general surgeons and otolaryngologists are associated with better survival outcomes than those performed by other specialties [15]. However, these results require validation in broader, real-world contexts, such as the US healthcare system. Moreover, the effect of surgical specialty must be interpreted within the context of other patient-level factors, including tumor stage and histology, comorbidities, medication use, and biochemical status at the time of surgery.

To address this knowledge gap, we conducted a large, real-world, retrospective cohort study using the TriNetX US Collaborative Network, a federated health research database comprising over 100 healthcare organizations. We primarily compared long-term survival between thyroidectomy and non-surgical management; secondarily, we conducted exploratory specialty-stratified analyses (ENT, GS, others). Using robust propensity score matching and multivariate adjustment, this study aims to describe whether outcomes vary across operating specialties in exploratory analyses, recognizing the potential for residual confounding. While clinical emphasis should remain on timely surgery within multidisciplinary pathways, our findings could inform referral practices and institutional credentialing; whether specialty should be prioritized requires prospective validation.

## 2. Materials and Methods

### 2.1. Data Source and Study Population

This retrospective cohort study utilized the TriNetX US Collaborative Network, a federated real-world data platform containing de-identified electronic health records (EHRs) from 105 healthcare organizations (HCOs) across the United States. Adults aged ≥20 years with a recorded diagnosis of malignant neoplasm of the thyroid gland (ICD-10-CM: C73) were eligible. The index date for both cohorts was defined as the date of the first inpatient encounter with a recorded C73 diagnosis.

Surgical cohort: Patients with a qualifying thyroid cancer C73 diagnosis who underwent thyroidectomy for malignancy, identified using CPT codes (60240, 60252, 60254, 60260, 60270, 60271) and corresponding ICD-10-PCS or SNOMED codes (including total, subtotal, completion thyroidectomy, and relevant neck lymphatic dissection procedures).

Non-surgical cohort (controls) comprised patients with a C73 diagnosis but without any thyroidectomy, completion thyroidectomy, neck dissection, laryngectomy, or tracheostomy procedure codes at any time.

Patients were excluded if they had incomplete demographic data, a prior thyroidectomy before the index date, death recorded before or on the index date, or <12 months of available EHR data prior to the index date. Follow-up began 1 day after the index date and continued until death or the last recorded clinical encounter.

### 2.2. Exposure and Outcomes

The exposure was the receipt of thyroid surgery. The primary outcome was all-cause mortality during follow-up (death recorded in the EHRs or linked mortality sources). Secondary analyses evaluated surgical-specialty-specific outcomes (otolaryngology [ENT], general surgery [GS], and others). Patients were followed from the index date until death or the last recorded clinical encounter.

### 2.3. Classification of Surgical Specialties

The surgical specialty was determined using provider-level metadata in TriNetX at the index thyroidectomy encounter: ENT—providers labeled otolaryngology or otolaryngology–head and neck surgery; GS—providers labeled general surgery or endocrine surgery; others—all remaining specialties (e.g., thoracic, plastic, oncology).

If a patient had multiple provider specialty labels in the same encounter, the main surgical specialty documented for the thyroidectomy was used. If conflict persisted, the surgeon’s specialty superseded the departmental labels [4,16]. Specialty-stratified outcomes were prespecified as secondary, exploratory analyses.

### 2.4. Covariates

Baseline covariates included the demographics (age, sex, race/ethnicity), comorbidities (hypertension, diabetes, metabolic syndrome, bone density disorders, hypothyroidism, hyperthyroidism, breast cancer, vocal cord paralysis, hypoparathyroidism, cardiovascular disease), selected procedures (oral radiopharmaceutical therapy), medication use (levothyroxine, platinum agents, monoclonal antibodies, immunotherapies), and laboratory values (TSH, FT4, serum calcium, creatinine, eGFR, albumin, hemoglobin). Comorbidities were identified using ICD-10-CM codes within 12 months prior to the index; the laboratory values were the closest pre-index measurement.

### 2.5. Propensity Score Matching

Propensity scores (PSs) were calculated using a logistic regression model that included all baseline covariates. One-to-one nearest-neighbor matching without replacement was performed with a caliper width of 0.2 × SD of the logit (PS). Covariate balance was assessed using standardized mean differences (SMDs); values < 0.1 indicated adequate balance. Before matching, the propensity score SMD was 0.248, indicating an imbalance. After matching, 49,219 patients were included in each cohort, and the PS SMD decreased to <0.02, where all baseline covariates achieved SMD < 0.1 (Table 1).

To evaluate the common support and matching quality, propensity score density plots were generated using TriNetX (Figure 1). Before matching (Figure 1), the surgery (purple) and control (green) cohorts had clearly separated distributions (mean PS = 0.61 vs. 0.43; SMD = 0.248). After matching (Figure 1), the distributions overlapped extensively (mean PS = 0.54 vs. 0.53; SMD < 0.02), indicating improved balance [17,18].

The proportion of patients within the caliper increased from 82% before matching to >99% after matching. These metrics and density functions align with recommended PS diagnostic procedures in observational studies [19].

### 2.6. Subgroup

The histologic subtype was abstracted from the site-contributed oncology registry or pathology-linked structured fields. Where available, International Classification of Diseases for Oncology, Third Edition (ICD-O-3) morphology codes were mapped into four prespecified categories: papillary, follicular, oncocytic (Hürthle cell), and medullary. When the ICD-O-3 morphology was unavailable, subtype assignments were cross-checked using ICD-10-CM thyroid cancer subtype codes and registry text fields via a prospectively defined crosswalk. The TNM components (T, N, M) and AJCC stage were captured from structured staging fields provided by participating sites. To align the staging with the treatment decision point, entries had to occur within −30 to +14 days of the index date.

### 2.7. Statistical Analysis

Continuous variables are expressed as the mean ± SD and categorical variables as the count (%). Between-group comparisons before matching used Student’s *t*-test and χ^2^ tests. Kaplan–Meier curves and log-rank tests compared the overall survival between the groups. Adjusted hazard ratios (HRs) and 95% confidence intervals (CIs) were calculated using Cox proportional hazard models, including subgroup analyses by surgical specialty. Sensitivity analyses used pairwise PSM. Analyses were performed using the TriNetX real-time analytics platform; *p* < 0.05 was considered statistically significant.

This study was approved by the Tungs’ Taichung MetroHarbor Hospital Institutional Review Board, where the ethical approval code is #113026.

## 3. Results

### 3.1. Baseline Characteristics After Matching

After the 1:1 propensity score matching, 49,219 patients were included in both the thyroid surgery and control cohorts. The two groups were well-balanced across all the baseline characteristics, with standardized differences (Std diff.) less than 0.1 for all the variables (Table 1).

The mean age at index was similar between the groups (51.6 ± 15.1 vs. 51.8 ± 15.3 years, Std diff. = 0.01). Female patients accounted for approximately 74% in both groups. The racial and ethnic distributions were comparable, although the proportion of unknown race was slightly higher in the thyroid surgery group (7.7% vs. 6.2%, Std diff. = 0.061), yet still within an acceptable balance.

The prevalences of comorbid conditions, including cardiovascular diseases (27.5% vs. 27.7%), osteoporosis (3.9% in both groups), hypothyroidism (11.2% vs. 11.1%), hyperthyroidism (3.3% vs. 3.1%), metabolic syndrome (0.3% in both), diabetes (9.4% vs. 9.5%), hypertension (21.1% vs. 21.4%), breast cancer (2.2% vs. 2.3%), and hypoparathyroidism (0.6% in both), showed no significant difference post-matching. Immune-related disorders and vocal cord paralysis were rare in both groups and remained balanced.

Regarding the procedures, oral radiopharmaceutical therapy and intra-articular injections occurred in <1% of patients and showed minimal differences. Use of medications, including levothyroxine (22.4% vs. 22.9%), platinum compounds, monoclonal antibodies, and immunotherapies (e.g., pembrolizumab), was well-matched across the groups.

Laboratory values were generally balanced, although small standardized differences were observed for TSH (Std diff. = 0.141), hemoglobin (0.157), serum albumin (0.133), and serum calcium (0.087), all of which were within the range of minor imbalance. Other parameters such as eGFR, free T4, and serum creatinine were well-matched (Std diff. < 0.1).

These balances support the primary comparison of thyroidectomy versus non-surgical care in subsequent survival analyses.

### 3.2. Kaplan–Meier Survival Curves

In this large-scale cohort study based on TrinetX with rigorous propensity score matching, patients who underwent thyroid surgery had a significantly improved overall survival compared with the matched controls without surgery. After the 1:1 PSM, 49,219 patients were included per cohort. Thyroidectomy was associated with lower long-term all-cause mortality compared with non-surgical care (adjusted HR 0.685, 95% CI 0.652–0.721), indicating a 31.5% relative reduction in the hazard of death among the surgical patients (Figure 2). The survival curves separated early and remained distinct throughout the follow-up.

The propensity score matching was performed using multiple key baseline covariates, including age at index date, sex, race/ethnicity (White, Black or African American, Asian, Hispanic or Latino, other race), comorbidities (hypertension, diabetes mellitus, metabolic syndrome, disorders of bone density, hypothyroidism, hyperthyroidism, malignancy of breast, vocal cord paralysis, hypoparathyroidism), selected procedures, medications (e.g., levothyroxine, targeted therapies), and laboratory parameters (e.g., thyrotropin, free thyroxine, calcium, creatinine, albumin, hemoglobin).

These findings underscore the potential survival benefit of thyroid surgery in appropriately selected patients and highlight the importance of controlling for multiple confounding factors through robust PSM.

### 3.3. Forest Plot Analysis

Our adjusted hazard ratios (HRs) in Figure 3 and pairwise propensity score-matched analysis in Figure 4 consistently demonstrated significant differences in overall survival between the patients that underwent thyroid surgery across different surgical specialty groups. Patients treated using general surgery (GS) showed superior survival outcomes, with a significantly lower risk of all-cause mortality, whereas those operated on by other specialties exhibited an increased risk compared with the otolaryngology (ENT) group. These findings emphasize that the surgical specialty may be associated with postoperative long-term all-cause mortality, underscoring the importance of specialized expertise and multidisciplinary consideration when planning thyroid cancer surgeries.

### 3.4. Subgroup Hazard Ratios for All-Cause Mortality

In the full cohort (Figure 5), thyroidectomy was associated with lower all-cause mortality across the TNM components (T: HR 0.53, 95% CI 0.48–0.59; N: 0.53, 0.48–0.59; M: 0.51, 0.45–0.57). By AJCC stage, the adjusted HRs were 0.62 (0.51–0.76) for stage I, 0.56 (0.42–0.73) for stage II, 0.61 (0.48–0.76) for stage III, and 0.45 (0.38–0.53) for stage IV (stage 0: 0.56, 0.23–1.31). By histology, the HRs were 0.74 (0.63–0.86) for papillary and 0.57 (0.34–0.95) for follicular carcinoma; estimates for oncocytic/Hürthle cell (0.63, 0.28–1.44) and medullary (0.52, 0.24–1.11) were imprecise, with the CIs crossing 1.

The findings were directionally consistent in the matched analyses. In the PSM cohort, the HRs were 0.61 (0.54–0.68) for T, 0.57 (0.51–0.65) for N, and 0.63 (0.55–0.72) for M. The stage-specific HRs were 0.69 (0.56–0.85) for stage I, 0.60 (0.45–0.81) for stage II, 0.70 (0.53–0.92) for stage III, and 0.55 (0.45–0.67) for stage IV; stage 0 was not estimable due to very low events. By histology, the HRs were 0.79 (0.67–0.94) for papillary and 0.41 (0.24–0.73) for follicular; oncocytic/Hürthle cell and medullary were not estimable in the matched sample. Overall, the magnitude and direction of effects in the PSM were similar to the full-cohort Cox estimates, supporting the association robustness.

## 4. Discussion

This study presents a comprehensive analysis of thyroid surgery outcomes, leveraging real-world data to compare both surgical versus non-surgical management and the impact of surgical specialty (ENT vs. GS) on long-term survival. Our results suggest a significant survival advantage in the thyroid cancer patients that underwent thyroidectomy, with secondary, exploratory specialty-stratified analyses indicating differences (Figure 3): compared with ENT (reference), GS showed a lower adjusted hazard, but these findings should be interpreted cautiously given potential residual confounding.

These findings align with and expand upon previous studies, which often emphasized short-term complications or disease-specific endpoints but lacked long-term survival data stratified by surgeon specialty. For example, Konuthula et al. observed higher complication rates in ENT-performed total thyroidectomies, possibly reflecting complex case selection patterns rather than surgical skill alone [20]. Ramos-Gonzalez et al. found no significant short-term differences in pediatric populations, though ENT tended to manage more complex cases [21]. Our data observed specialty-related differences in exploratory analyses, yet causal implications for long-term outcomes cannot be determined from these observational comparisons. Prior reports often focused on short-term complications or distinct populations, limiting the direct applicability to long-term survival in adult malignant disease.

Although ENTs are anatomically specialized, our exploratory analyses observed lower adjusted hazards with GS; the case mix, surgeon/center volume, and institutional pathways could underlie this observation, as prior studies show that GSs tend to perform more standardized resections with potentially lower intraoperative variability [5,22]. Moreover, Al-Qahtani et al. reported that while ENT surgeons more frequently utilized intraoperative neuromonitoring and general/endocrine surgeons had shorter operative times, immediate complications were not consistently fewer in the GS group (transient vocal cord palsy occurred only in that group), limiting inferences about longer-term recovery trajectories [23].

Beyond the specialty, our study adds to growing evidence supporting the survival benefit of thyroid surgery itself. Many earlier studies focused on short-term complications, such as hypocalcemia, recurrent laryngeal nerve injury, and infection [24,25,26], while only a few explored survival endpoints [27,28]. Our data, reflecting a ~31.5% reduction in all-cause mortality for surgical vs. non-surgical management, are consistent with and add to prior work suggesting that surgical intervention in appropriately selected patients may confer systemic survival advantages [29,30,31].

This supports recent discussions advocating for a more balanced approach to thyroid disease management. While guidelines increasingly support conservative management, especially for low-risk nodules or differentiated thyroid cancers [32,33,34], our data suggest that timely surgery within multidisciplinary pathways remains central when appropriate, with patient selection and shared decision-making as key determinants. These findings are consistent with population-level studies demonstrating decreased thyroid cancer mortality among surgically treated patients [35,36,37].

Our use of PSM improved the covariate balance and enhanced the internal validity of the associations but could not eliminate unmeasured confounders or establish causality. Unlike many prior retrospective analyses [38,39,40], we achieved balance across demographic, clinical, and laboratory variables, supporting causal inference. This methodological rigor mirrors the strengths of other robust EHR-based studies, such as those utilizing SEER or ACS-NSQIP datasets [41,42,43].

Furthermore, our findings suggest that outcomes may vary by specialty and warrant investigation into underlying drivers, such as the case mix, volume, and institutional pathways. While ENT surgeons may handle more anatomically complex or oncologically challenging cases [20,21,22,23], this should prompt evaluation of differences in case selection, staging workflows, use of standardized pathways, and surgeon/center volume across specialties, identifying quality-improvement opportunities for all groups. Future research should explore whether ENT surgical outcomes can be improved through institutional standardization, collaborative decision-making, or joint surgical models [7,44,45].

The study may motivate evaluation of referral pathways; institutions may evaluate assignment patterns considering local volume and expertise, pathway adherence, and multidisciplinary coordination [5,28]. Multidisciplinary collaboration, including tumor boards and shared case planning, may further optimize outcomes regardless of specialty background [46,47,48].

Incomplete capture of the TNM stage and histology across sites may introduce residual confounding despite the use of predefined proxies and subgroup reporting. Misclassification of surgical specialty is possible due to the reliance on provider metadata, although sensitivity checks (e.g., cross-checking with procedure types, excluding low-volume providers) yielded directionally similar estimates. As with all observational studies, unmeasured confounding cannot be fully excluded. Additionally, surgeon-specific experience, volume, and institutional protocols may have influenced the outcomes but were not directly measurable. Nonetheless, the large matched cohort, national representativeness, and robust statistical approach strengthened the generalizability of our findings.

## 5. Conclusions

In this large, propensity-matched, real-world cohort, thyroidectomy was associated with lower long-term all-cause mortality than non-surgical management. In secondary, exploratory specialty-stratified analyses, GS showed a lower adjusted hazard of death than ENT, whereas other specialties showed higher hazards; these observations are hypothesis-generating and may reflect residual confounding (e.g., unmeasured tumor stage and histologic risk), referral patterns, surgeon or center experience, and possible specialty misclassification. Overall, these findings support timely surgery within multidisciplinary pathways and may inform local referral and credentialing discussions, while prospective studies with tumor-level annotation and provider/center characteristics are needed to clarify causality and the extent of specialty-related variation.

## Figures and Tables

**Figure 1 cancers-17-03051-f001:**
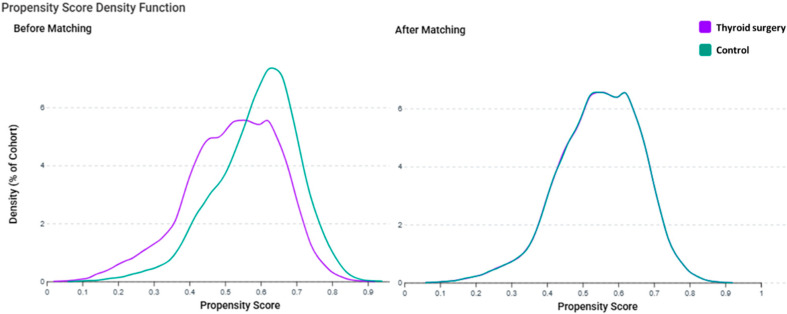
Propensity score density functions before and after matching. Distribution of propensity scores for the thyroid surgery (purple) and control (green) cohorts before matching (**left**) and after matching (**right**). After matching, the two groups showed substantial overlap, indicating an improved covariate balance.

**Figure 2 cancers-17-03051-f002:**
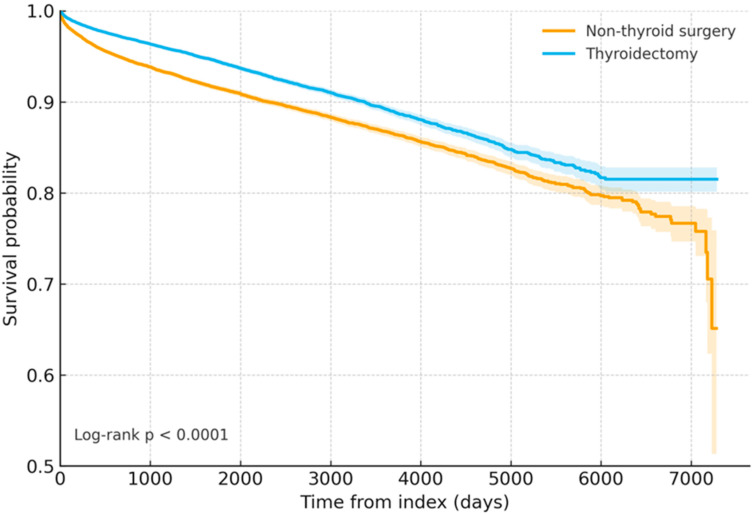
Kaplan–Meier survival curves comparing case patients that underwent thyroid surgery and matched controls without thyroid surgery after propensity score matching (PSM). After the PSM, each group included 49,219 patients. The case group demonstrated a significantly higher survival probability during the follow-up period. The log-rank test indicated a significant difference between the two groups (χ^2^ = 217.614, *p* < 0.0001). The adjusted hazard ratio (HR) for overall survival was 0.685 (95% confidence interval, 0.652–0.721), favoring the surgery group.

**Figure 3 cancers-17-03051-f003:**
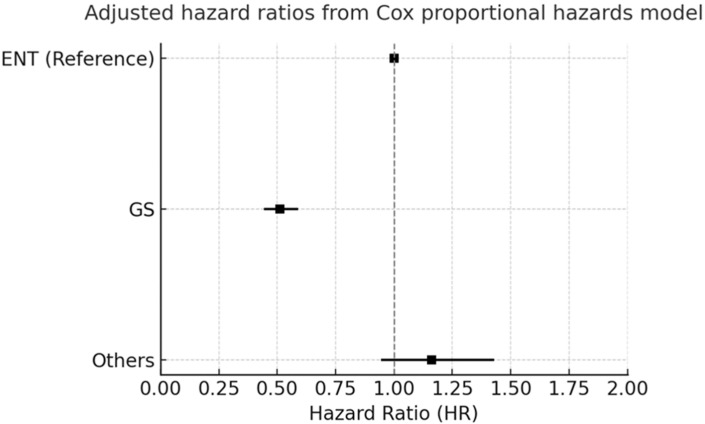
Adjusted hazard ratios (HRs) for overall survival among the patients that underwent thyroid surgery by surgical specialty group. The forest plot depicts HRs derived from a multivariate Cox proportional hazards model that compared general surgery (GS) and other specialties (others) with otolaryngology (ENT) as the reference group. Squares indicate point estimates of HRs, and horizontal lines represent 95% confidence intervals.

**Figure 4 cancers-17-03051-f004:**
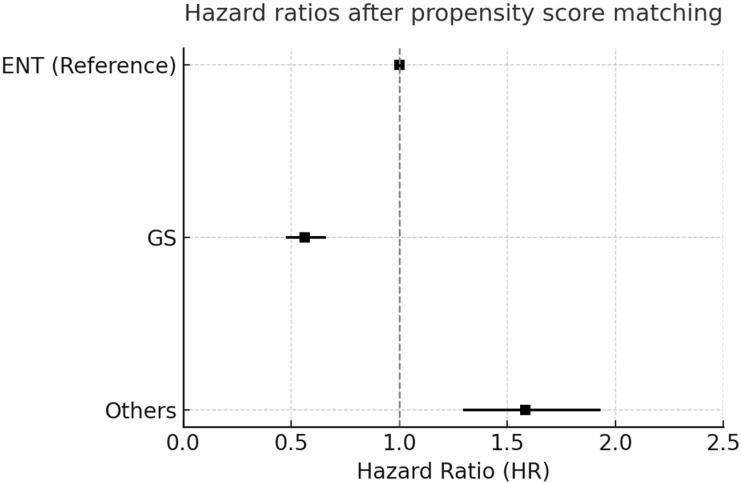
Pairwise propensity score-matched hazard ratios (HRs) for overall survival among patients that underwent thyroid surgery by surgical specialty group. Forest plot illustrating hazard ratios (HRs) and 95% confidence intervals (CIs) for overall survival derived from pairwise propensity score-matched analyses that compared different surgical specialty groups. Compared with the ENT group, the GS group showed a significantly lower risk (HR = 0.561; 95% CI: 0.481–0.654), whereas the others group exhibited a higher risk (HR = 1.583; 95% CI: 1.302–1.924).

**Figure 5 cancers-17-03051-f005:**
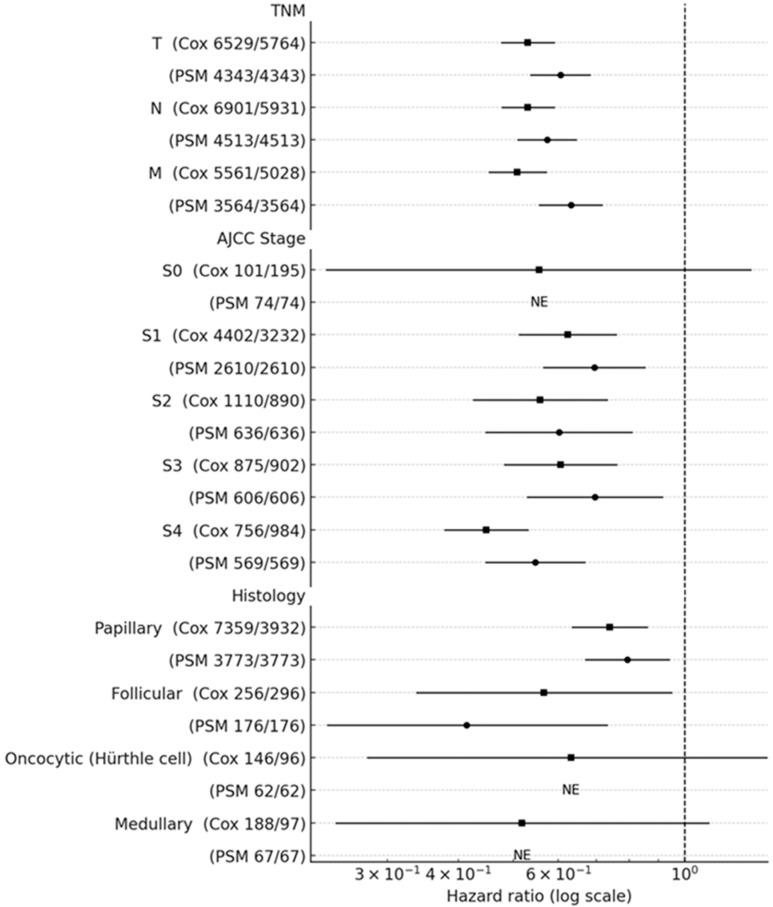
Subgroup hazard ratios for all-cause mortality comparing thyroidectomy vs. non-thyroid surgery. Black squares denote adjusted hazard ratios (HRs) from the multivariable Cox model in the full cohort; black circles denote HRs from the 1:1 propensity score-matched (PSM) cohort. Horizontal bars show 95% confidence intervals on a log scale; the vertical dashed line marks HR = 1. Subgroups are ordered as TNM components (T, N, M), AJCC stage (S0–S4), and histology (papillary, follicular, oncocytic [Hürthle cell], medullary). Labels include the number of patients that underwent the surgery and control groups for each row. For matched strata with non-estimable effects due to zero or very few events, NE is shown at the corresponding Cox HR to facilitate a visual comparison. Values < 1 favor thyroidectomy.

**Table 1 cancers-17-03051-t001:** Basic characteristics before propensity score matching.

	Thyroid Surgery		Control		*p*-Value	Std Diff.
	(N = 49,219)		(N = 49,219)			
	Patients	%	Patients	%		
Demographics						
Age at index (Mean ± SD)	51.6 ± 15.1		51.8 ± 15.3		0.125	0.01
Female	36,507	74.20%	36,430	74.00%	0.575	0.004
Male	12,712	25.80%	12,789	26.00%	0.575	0.004
Race						
White	28,906	58.70%	29,502	59.90%	<0.001	0.025
Black or African American	3224	6.60%	3169	6.40%	0.477	0.005
Asian	3591	7.30%	3683	7.50%	0.262	0.007
Unknown race	3791	7.70%	3030	6.20%	<0.001	0.061
Ethnicity						
Hispanic or Latino	3291	6.70%	3317	6.70%	0.741	0.002
Not Hispanic or Latino	27,523	55.90%	27,528	55.90%	0.974	<0.001
Unknown ethnicity	18,405	37.40%	18,374	37.30%	0.838	0.001
Diagnosis						
Cardiovascular diseases	13,516	27.50%	13,654	27.70%	0.325	0.006
Osteoporosis or bone disorders	1902	3.90%	1911	3.90%	0.882	0.001
Hypothyroidism	5493	11.20%	5463	11.10%	0.761	0.002
Hyperthyroidism	1600	3.30%	1536	3.10%	0.245	0.007
Metabolic syndrome	143	0.30%	142	0.30%	0.953	<0.001
Diabetes	4649	9.40%	4697	9.50%	0.602	0.003
Immune-related disorders	49	0.10%	52	0.10%	0.765	0.002
Hypertension	10,397	21.10%	10,555	21.40%	0.219	0.008
Breast cancer	1102	2.20%	1128	2.30%	0.578	0.004
Vocal cord paralysis	740	1.50%	708	1.40%	0.397	0.005
Hypoparathyroidism	304	0.60%	282	0.60%	0.362	0.006
Procedure						
Oral radiopharmaceutical therapy	278	0.60%	276	0.60%	0.932	0.001
Intra-articular injection (percutaneous)	20	0.00%	18	0.00%	0.746	0.002
Medication						
Levothyroxine	11,030	22.40%	11,284	22.90%	0.053	0.012
Sorafenib	10	0.00%	14	0.00%	0.414	0.005
Cabozantinib	10	0.00%	12	0.00%	0.670	0.003
Vandetanib	10	0.00%	10	0.00%	1.000	<0.001
Platinum compounds	279	0.60%	273	0.60%	0.798	0.002
Monoclonal antibodies	332	0.70%	345	0.70%	0.616	0.003
Methylhydrazines	10	0.00%	10	0.00%	1.000	<0.001
Pembrolizumab	67	0.10%	71	0.10%	0.733	0.002
Laboratory (Mean ± SD)						
TSH (thyroid stimulating hormone)	4.2 ± 24.2		8.3 ± 33.3		<0.001	0.141
Free T4 (free thyroxine)	1.3 ± 0.8		1.3 ± 0.7		<0.001	0.074
Serum calcium	9.3 ± 0.8		9.2 ± 0.9		<0.001	0.087
Serum creatinine	1.0 ± 2.0		1.0 ± 2.2		0.109	0.015
eGFR (estimated GFR, MDRD)	83.2 ± 26.6		82.2 ± 29.2		<0.001	0.036
Serum albumin	4.1 ± 0.5		4.1 ± 0.5		<0.001	0.133
Hemoglobin	13.2 ± 1.8		13.0 ± 1.9		<0.001	0.157

Continuous variables are expressed as mean ± standard deviation (SD), and categorical variables as number and percentage. Comparisons between groups were conducted using Student’s *t*-test for continuous variables and chi-square tests for categorical variables. Covariate balance was assessed using standardized differences (Std diff.), with values <0.1 indicating negligible imbalance.

## Data Availability

The data that support the findings of this study are available from the TriNetX US Collaborative Network (https://trinetx.com, accessed on 10 July 2025). Restrictions apply to the availability of these data, which were used under license for this study and are therefore not publicly available.

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
