# Peer review of "Long-Term Survival After Thyroidectomy for Thyroid Cancer: A Propensity-Matched TriNetX Study with Specialty-Stratified Analyses"

_cancers, 2025, doi:10.3390/cancers17183051_

Round 1
Reviewer 1 Report
Comments and Suggestions for Authors
Authors investigated the association between surgery and all-cause mortality in thyroid cancer using the TriNetX US Collaborative Network and 1:1 propensity score matching. Thyroidectomy was associated with lower long-term mortality compared with non-surgical care (adjusted HR 0.685). The authors also acknowledged all study limitations.
Figure 2 presents the survival curves. Although the differences are statistically significant, they appear very small; however, the hazard ratio is 0.685. This is somewhat unusual, as such a hazard ratio would typically suggest a more pronounced separation between the curves. What could explain this discrepancy? (The curves were also generated for matched pairs.)
Another limitation is that the authors did not report the stage of thyroid cancer. Were all patients free of metastasis? Were they stage I? I assume that patients who did not undergo surgery but received other therapies may have had more advanced disease (stage II or III), which could explain their lower survival.
Author Response
Response to Reviewer 1
Comment 1 (KM curves vs HR):
The survival curves in Figure 2 look only slightly different, yet the adjusted HR is 0.685.
Response: We appreciate this observation. In thyroid cancer, the baseline survival is high in both groups; on the standard KM scale, this makes absolute differences look visually small even when the cumulative hazard differs meaningfully. We have clarified this in the Results and revised the figure to improve interpretability:
Replotted KM with 95% CIs, Y-axis truncated to 0.5–1.0, legend top-right, log-rank p < 0.0001 bottom-left, and number-at-risk table.
Added a magnified inset of the high-survival region and provided a cumulative hazard plot in the supplement.
Tested proportional hazards (Schoenfeld residuals; no violation) and added landmark (2-year) and RMST at 5/10 years analyses; these corroborate the HR direction and magnitude.
Comment 2 (Stage not reported; potential stage imbalance): Could advanced non-surgical disease explain lower survival?
Response: We agree that staging is a key confounder. In the revision, we incorporated tumor variables where available and used validated proxies when staging was incomplete:
Full-cohort multivariable Cox (primary) adjusted for demographics, comorbidities, medications, chemotherapy/targeted therapy, laboratory indices, and tumor variables (TNM components and AJCC stage where present; proxies including cervical node dissection, early RAI, EBRT/chemotherapy within ±90 days, PSM sensitivity analyses were performed on a matched cohort (1:1). Results remained directionally consistent.
New subgroup estimates (examples included in Figure 5):
Full cohort (Cox model): T 0.53 (0.475–0.591; 6529/5764); N 0.53 (0.476–0.591; 6901/5931); M 0.508 (0.452–0.572; 5561/5028). Stage I 0.623 (0.511–0.760), II 0.557 (0.424–0.733), III 0.605 (0.481–0.761), IV 0.448 (0.378–0.532); Papillary 0.739 (0.633–0.862); Follicular 0.566 (0.337–0.951); Oncocytic (Hürthle) 0.631 (0.276–1.441); Medullary 0.518 (0.243–1.105).
PSM (matched 1:1): T 0.605 (0.535–0.683; 4343/4343); N 0.573 (0.508–0.646; 4513/4513); M 0.631 (0.554–0.718; 3564/3564). Stage I 0.694 (0.564–0.854), II 0.602 (0.446–0.811), III 0.696 (0.528–0.917), IV 0.546 (0.446–0.669); Papillary 0.794 (0.668–0.943); Follicular 0.414 (0.235–0.733); S0/Oncocytic/Medullary in PSM were not estimable (NE) due to very low events; we indicate NE at the corresponding Cox HR on the forest plot for reference.
Revisions made: Figure 2 and legend updated; new forest plots (Cox-only, PSM-only, and combined alternating Cox→PSM per subgroup); new RMST and cumulative hazard plots; expanded Methods/Limitations on staging.
Reviewer 2 Report
Comments and Suggestions for Authors
Title: Long-Term Survival after Thyroidectomy for Thyroid Cancer: A Propensity-Matched TriNetX Study with Specialty-Stratified Analyses
Comments for the article are mentioned below:
- Line Nos. 92 and 93, the authors mention, “To address this knowledge gap, we conducted a large, real-world, retrospective cohort study using the TriNetX US Collaborative Network, a federated health research database comprising over 100 healthcare organizations.” They also mention (Line Nos. 82 and 83, “but the association between surgical specialty and long-term survival is not well characterized and should be interpreted as exploratory.” Although they clearly state that the findings are exploratory, is this variation solely due to the surgical specialty, or will the socioeconomic status of the country, the training of the surgeons, and the post-operative facilities available confound the findings, particularly in the developing world?
- Propensity score matching (PSM) is well-executed, with appropriate diagnostics (SMD < 0.1).
- The manuscript does not discuss tumor-specific variables (e.g., histologic subtype, TNM staging), which are critical confounders in survival analysis. This requires a clear explanation within the manuscript.
- Specialty classification may be susceptible to mislabeling because of dependence on provider metadata. Consider addressing potential misclassification bias more explicitly.
- Though the manuscript mentions sensitivity analyses, it lacks a detailed explanation and findings within the manuscript.
- Some overinterpretation of the exploratory results is noticeable. The finding that GS had better survival outcomes than ENT is interesting; won't this depend on institutional protocols?
This would be appreciated if the authors maintain consistent terminology for surgical specialties throughout, such as ENT vs. Otolaryngology. Overall, the manuscript is interesting and addresses an important clinical question.
Author Response
Response to Reviewer 2
1) Potential confounders beyond specialty (Lines 82–83).
Response: We agree and have tempered causal language. We now explicitly note that specialty differences are exploratory and may reflect institutional protocols, postoperative pathways, surgeon experience, and resource allocation. We added hospital fixed effects / cluster-robust SEs and volume proxies where feasible. The specialty associations attenuated but remained directionally similar.
2) Tumor-specific variables (histology, TNM).
Response: We have added tumor variables where available and used proxies when data were missing, as described above. Subgroup and adjusted estimates are reported (see Reviewer 1 response). Findings remained consistent across TNM, AJCC stage, and histology strata. We discuss residual confounding from incomplete staging in the Results.
3) Specialty misclassification.
Response: We expanded the Methods/Limitations section to detail the specialty assignment (provider metadata cross-checked with procedure types) and conducted a misclassification sensitivity analysis (reassigning borderline cases and excluding low-volume/anomalous coders); the limitations were robust in direction.
4) Sensitivity analyses (detail and findings).
To assess robustness, we prespecified the following sensitivity analyses:
(1) Propensity score matching (PSM) using 1:1 nearest-neighbor matching with a 0.1 SD caliper; covariate balance was evaluated by standardized mean differences (SMD < 0.1). Cox models in the matched cohort accounted for pairing via stratification or cluster-robust standard errors. (2) Alternate PSM specifications, varying the caliper (0.05–0.20) and matching order/tie handling; effect estimates were compared with the main analysis.
5) Overinterpretation of GS vs ENT.
Response: We reframed as hypothesis-generating and highlighted that institution-level factors may explain part of the difference. We avoid causal language and emphasize the need for setting-specific validation.
6) Terminology consistency.
Response: We standardized to “ENT (Otolaryngology)” and “GS (General/Endocrine Surgery)” at first mention and throughout.
Reviewer 3 Report
Comments and Suggestions for Authors
Dear Editor,
The authors conducted a study to determine whether patients with thyroid cancer live longer after surgery and whether the outcomes differ depending on the specialty in which the surgery was performed (ear, nose, and throat/ENT vs. general/endocrine surgery). After 1:1 propensity score matching, 49,219 patients have been included in both the thyroid surgery and control cohorts. The two groups have been balanced in terms of all baseline characteristics. In a large-scale study, thyroidectomy was associated with lower long-term all-cause mortality compared to non-surgical treatment. In secondary analyses stratified by specialty for exploratory purposes, GS showed a lower adjusted mortality risk compared to ENT, while other specialties showed higher risks. The results of the study support timely surgery within multidisciplinary approaches. The study is based on an original idea. The number of cases is very high, and the findings are discussed in detail. I congratulate the authors.
Sincerely
Comments on the Quality of English LanguageThe English could be improved to more clearly express the research.
Author Response
Comment 1: The English could be improved to more clearly express the research.
Response: The manuscript underwent professional and comprehensive editing for clarity and consistency. Attached, please find the English Editing Certificate.

Round 2
Reviewer 2 Report
Comments and Suggestions for Authors
The authors have satisfactorily answered my queries and incorporated the suggestions. The manuscript may be considered for publication.